# Addressing Data Heterogeneity Through a Pre-learned Manifold for Distributed Learning Scenarios

## Abstract

In distributed learning environments like federated learning, data heterogeneity across clients has been a key challenge, which often leads to suboptimal model performance and convergence issues. So far, plenty of efforts have focused on addressing data heterogeneity by relying on a hypothetical clustering structure or a consistent information-sharing mechanism. However, because of the inherent complexity and diversity of real-world data, these assumptions may be largely violated. In this work, we argue that information sharing is mostly fragmented in the collaboration network in reality. The distribution overlaps are not consistent but scattered among local clients. We propose the concept of *Precision Collaboration*, which refers to accurately identifying the informative data in other clients precisely while carefully avoiding the potential negative transfer induced by others. In particular, we propose to pre-learn a global manifold, which infers the local data manifolds and estimates the exact local data density simultaneously. The learned manifold aims to precisely identify the shared data in other clients. The estimated exact likelihood allows for generating samples from the manifold precisely. Our pre-training strategy enables reusable and scalable model learning, especially when an ongoing influx of new clients becomes part of the network. Experiments show that our proposed method effectively identifies the favorable data in other clients without compromising privacy preservation, and significantly overcomes baselines on benchmarks and a real-world clinical data set.

## 1 Introduction

Distributed learning frameworks, such as federated learning (FL), have garnered significant attention across various fields in recent years (Wen et al., 2023). It enables collaborative model learning when training data are collected by multiple clients in a network. As it learns from distributed data sources without the need to access the raw data across different clients, it facilitates real-world scenarios where privacy preservation is crucial, such as finance (Yang et al., 2019), healthcare (Xu et al., 2021) and criminal justice (Berk, 2012). While it is common that the data samples in local clients are non-i.i.d., existing research reveals that ***data heterogeneity*** could lead to non-guaranteed convergence, inconsistent performance and catastrophic forgetting across local clients (Qu et al., 2022). Despite the promise of FL, an increasing concern is how to effectively handle data heterogeneity before FL is applied in real-world data scenarios.

To address this issue, personalization has emerged as a critical research direction. A variety of efforts have been made to explore this direction. For example, Ghosh et al. (2020) proposed to cluster the clients according to their sample distributions and build a customized model for each cluster. However, their hypothesis excludes the possibility of knowledge transfer across clusters. Li et al. (2021b) enhanced personalized model learning by introducing a global regularization term, which assumed that the shared knowledge was consistent across all clients.

Given the variety of local data distributions, we explore a more adaptable and generalized scenario where distribution overlaps may be fragmented, as illustrated in Figure 1 (a). Since the informative and ambiguous data shards exist simultaneously in other clients, collaborating with all data may do

harm to the model learning. An interesting and challenging problem is how to selectively collaborate with the favorable part of other clients.

In this paper, we put forward the concept "Precision Collaboration" for fragmented information sharing. To begin with, we argue that data heterogeneity comes from inconsistent local data manifolds, and the local data manifolds share different overlaps. Maximizing the benefit of collaboration requires a precise utilization of these overlaps. Moreover, local data are usually collected from the manifold based on a particular density. If we want to generate data from the manifold, a precise distribution density approximation for each client could facilitate model learning.

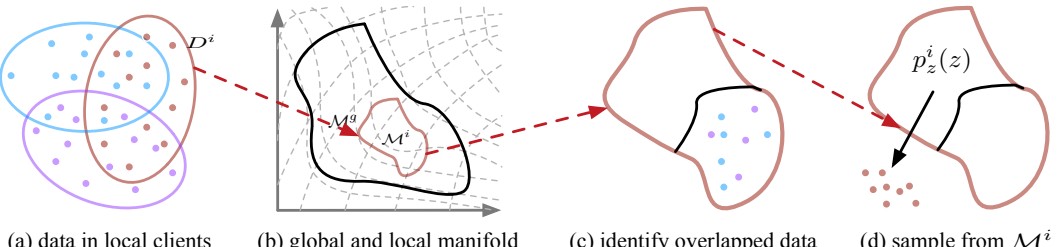

| (a) data in local clients | (b) global and local manifold | (c) identify overlapped data | (d) sample from $\mathcal{M}^i$ |

Figure 1: Overview of our proposed PCML. (a) Fragmented distribution overlaps exist among clients; (b) learn the global data manifold $\mathcal{M}^g$ and determine the local manifold $\mathcal{M}^i$ for each client; (c) the data from other clients lie on the local manifold $\mathcal{M}^i$ are identified as data overlaps; (d) learn a local density $p_z^i(z)$ for local data generation.

To realize our proposed precision collaboration, we develop a novel framework to achieve Precision Collaboration for Model Learning named PCML shown in Figure 1. We assert that the key to precisely collaborative model learning is identifying the distribution overlaps scattered in other clients and the local distribution density. To infer the local data manifold to identify the overlaps and approximate the local distribution simultaneously, we propose to learn a normalizing flow (NF) for the federated network. For the manifold learning, we first infer the underlying manifold $\mathcal{M}^g$ of the data from all clients by learning a global NF model. In this way, the data from all clients is utilized for manifold inference. Then the local manifold $\mathcal{M}^i \subset \mathcal{M}^g$ of the $i$-th client could be determined by local data $D^i$ as shown in Figure 1 (b).

From Figure 1 (c), the local data manifold $\mathcal{M}^i$ is used to identify the beneficial overlaps from other clients. In particular, if a subset of the data from $D^j$ lies on $\mathcal{M}^i$, this subset is the overlaps between the $i$-th and $j$-th clients. For the local distribution density learning, we approximate the local distribution density during the learning of the normalizing flow. We suggest sampling from $\mathcal{M}^i$ with the approximated local distribution density as shown in Figure 1 (d), which effectively mitigates the potential distribution discrepancy. During the model learning, we keep the raw data and the invertible representations learned by NF in local clients to avoid additional privacy leakage. We highlight our key contributions as follows:

- We propose the concept "Precision Collaboration", which refers to an optimal collaboration that learns from the inconsistently shared data shards and excludes the ambiguous information;

- We develop PCML to achieve precise model collaboration by pre-learning a data manifold. It enables a reusable and scalable model learning, especially as an ongoing influx of new clients joins the collaboration network;

- Empirical experiments corroborate that PCML significantly outperforms all baselines on a series of benchmark data sets and a real-world clinical data set.

## 2 RELATED WORK

### 2.1 DISTRIBUTED LEARNING SCENARIOS AND DATA HETEROGENEITY

Recent years have witnessed growing attention to distributed learning scenarios, e.g., federated learning (McMahan et al., 2017), of which several challenges have been concerning topics including

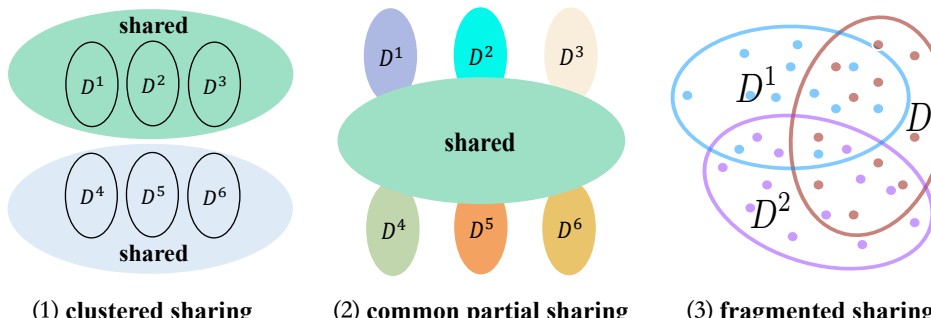

(1) **clustered sharing**    (2) **common partial sharing**    (3) **fragmented sharing**

Figure 2: Illustrations of the three assumptions on data heterogeneity. (a) **clustered sharing**; 1. all information is shared within the clusters; 2. the sharing is not consistent across all clients; (b) **common partial sharing**; 1. partial information within clients are shared; 2. the sharing is consistent across all clients; (c) **fragmented sharing**; 1. partial information within clients may be shared with others, 2. the sharing is not consistent.

communication efficiency (Konečný et al., 2016), privacy (Agarwal et al., 2018) and data heterogeneity (Karimireddy et al., 2020). While data heterogeneity could cause the lack of convergence and the potential of catastrophic forgetting (Qu et al., 2022), there are researchers aiming to tackle the heterogeneity by learning a global model. For example, Li et al. (2020) add a proximal term to constrain local updates, Mohri et al. (2019) maximize performance on arbitrary distributions, and Li et al. (2021a) use MOON to align local and global representations. Instead of pursuing a balanced performance distribution, we are interested in achieving the best performance for each client by precisely learning the shared informative overlaps from others. Cheng et al. (2024) introduce momentum to FedAvg and SCAFFOLD, boosting convergence and performance in non-IID settings without bounded heterogeneity assumptions.

## 2.2 PERSONALIZATION

In addition to reaching global consensus, personalized model learning also attracts widespread concern in FL community, which may boost the flexibility of learned models when adapting to local distributions Cui et al. (2022); Li et al. (2021b); Mclaughlin & Su (2024); Ghari & Shen (2024). Plentiful research has proposed techniques for a trade-off between local and global models. For example, Fallah et al. (2020) proposed to train local models that can quickly adapt to local data starting from an initial shared model in a meta-learning way. Some works train personalized models by interpolating between global and local models (Deng et al., 2020; Dinh et al., 2020). Li et al. (2021b) achieve such a trade-off through regularizing local models close to the global model. Shen et al. (2024) enhances generic federated learning by dynamically selecting personalized plug-in modules to optimize performance across diverse client data distributions. There are other works suggesting a partially shared model structure for efficient information transferring (Liang et al., 2020; Collins et al., 2021). Nonetheless, we argue that global models struggle to capture varied shared information, necessitating the precise identification of fragmented knowledge in collaborative learning.

To mitigate the potential overfitting when learning from limited local data, some works use generative methods to improve the model performance (Du & Wu, 2020; Zhu et al., 2021). Zhu et al. (2021) regulate local training with the distilled knowledge from all clients. Du & Wu (2020) lead into GAN for generating similar data for local clients. However, generating data at an arbitrary density could result in distribution discrepancy. An optimal sampling density may present more benefits for local learning tasks.

## 3 NOTATIONS AND PROBLEM DEFINITION

### 3.1 NOTATIONS

Suppose there are $N$ clients in a collaboration network, each client owns a private dataset $D^k$ with $n^k$ data samples. The dataset $D^k = \{X^k, Y^k\}$ consists of the input space $X^k$ and output space $Y^k$. We use $z = \{x, y\}$ to denote a data point, and $z \in \mathcal{M}$ denotes the data manifold. The input space

and the output space are shared across all clients. In the following, we also use $D^i$ to denote the $i$-th client without causing further confusion.

**Averaging Method.** The goal of each client is to learn the best model to predict the label $y$ by collaborating with others. For example, McMahan et al. (2017) propose `FedAvg`, which learns a global model $f$ for all clients by minimizing the empirical risk over the samples from all clients, i.e.,

$$\min_{f \in \mathcal{F}} \frac{1}{\sum_{k=1}^{N} n^k} \sum_{k=1}^{N} \sum_{i=1}^{n^k} l\left(f\left(x_i^k\right), y_i^k\right), \qquad (1)$$

where $\mathcal{F}$ is the hypothesis space and $l$ denotes the loss objective of all clients. From Eq.1, `FedAvg` assumes that the i.i.d. data from different clients associate with a common data manifold $\mathcal{M}$ and sampling density $p_z^g(z)$.

## 3.2 Assumptions on Data Heterogeneity

However, the i.i.d. assumption is largely violated as the local data distributions may be significantly distinctive Mohri et al. (2019). In this event, learning a consensus by averaging the local gradients could cause severe performance degradation on certain clients (Li et al., 2019b; Cui et al., 2021). There are research studying distributed learning scenarios with non-i.i.d. data, and the assumptions on data heterogeneity are mainly from the two perspectives.

**Clustered sharing.** As shown in Figure 2 (a), the clients partitioned in each cluster own common data manifolds ($\mathcal{M}^j$) and sampling density ($p_z^j(z)$). Clustered sharing requires that all message is shared within the clusters, and there is no knowledge transfer across clusters.

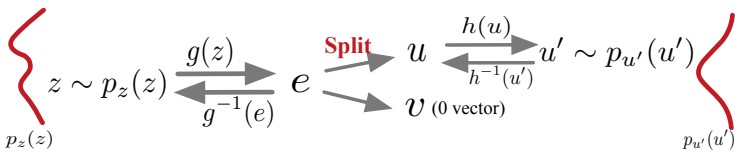

Figure 3: Illustration of the manifold learning via an NF method. By **Split** operation, the NF model could learn a low-dimensional latent representation $u'$, and the model parameters significantly decrease. For a complex distribution $p_z(z)$, the NF model learns a tractable injective chart $g \circ h$, which models $p_z(z)$ to a simple distribution $p_{u'}(u')$.

**Common partial sharing.** From Figure 2 (b), a common distribution overlap is shared across all clients. Meanwhile, each client owns specific knowledge that cannot be leveraged by others. This means that each client is associated with a specific data manifold $\mathcal{M}^i$, and the overlapped region of the manifold is shared across all clients.

Compared with the previous assumptions above, we study a more general scenario *fragmented sharing*, where the shared distribution overlaps are scattered among the clients. Besides, these overlaps are inconsistent across all clients as shown in Figure 2 (c).

**Fragmented sharing.** The local data $z \in D^i$ are sampled from the local manifold $\mathcal{M}^i$ in a particular density $p_z^i(z)$, and there exist overlaps among data manifolds, i.e.,

$$z \in \mathcal{M}^i \subset \mathbb{R}^d, \quad z \sim p_z^i(z) \qquad (2a)$$

$$\exists\, i, j \in \{0, 1, ..., N-1\}, \; s.t., \; \mathcal{M}^i \cap \mathcal{M}^j \neq \emptyset, \qquad (2b)$$

where $d$ in Eq.(2a) is the dimension of $z$. Eq.(2b) implies the data sharing a common distribution may not be consistent across all clients.

## 4 Methodology

### 4.1 Preliminaries: Normalizing Flow and Manifold Learning

**Normalizing flow (NF).** The generative method NF achieves exact likelihood estimation through an invertible transformation from a known distribution to a complex target distribution. Given a target

dataset $D = \{z_0, z_1, ..., z_{n-1}\}, z_i \in \mathbb{R}^d$ and a base variable $e \in \mathbb{R}^d$ with a known density $p_e(e)$, classic NF methods learn a diffeomorphism $f : e_i = g(z_i)$ which maps $p_z$ to the density $p_e$:

$$p_z(z) = p_e\left(g(z)\right) \left|\det J_g\left(g(z)\right)\right|^{-1}, \tag{3}$$

where $\det J_g\left(g(z)\right) \in \mathbb{R}^{d \times d}$ denotes the Jacobian matrix evaluated at $g(z)$. Since $g$ is bijective, it is trackable and Eq.(3) could be effectively computed. By fitting the dataset $D$, the approximated distribution $p'_z(z)$ is optimized through a pushforward operation. To enhance the scalability of $g$, one could compose several diffeomorphisms $g = g_{n-1} \circ \cdots \circ g_1 \circ g_0$ for a larger model capacity.

**Manifold learning via a NF model.** Consider a data generation procedure $z \in \mathcal{M} \subset \mathbb{R}^d$, where $\mathcal{M}$ is the data manifold embeded in a $d'$ dimensional latent space $d' < d$. A classical NF model requires a fixed dimensionality of the latent space, which is the same as the dimension of the data $d$. To approximate the low-dimensional manifold via NF, existing research Brehmer & Cranmer (2020) proposes to split the latent space shown in Figure 3. In particular, Given the data $z$, a bijective transformation $g_\theta$ is used to obtain the latent representation $e \in E$,

$$e = g_\theta(z), \quad where \quad z = g_\theta^{-1}(e). \tag{4}$$

Then the latent space $E$ is separated $E = U \times V$ as shown in Figure 3, where $U = \mathbb{R}^{d'}$ denotes the coordinates on the manifold. $V = \Bbbk^{d-d'}$ denotes the remaining coordinates, which are the directions orthogonal to the manifold. The separation operation improves the optimization efficiency because of fewer parameter dimensions, especially when the data $z$ is high-dimensional.

To model the density $p_u(u)$, the variable $u$ is transformed to the variable $u'$ with the given density $p_{u'}(u')$ using a bijective model $h_\phi$:

$$u' = h_\phi(u), \quad where \quad u = \text{Split}(e), \tag{5}$$

where $\text{Split}(e)$ denotes deleting the $d - d'$ dimensional $\Bbbk$ vector from $e$, and $\text{Pad}(u)$ denotes the inverse operation.

### 4.2 AN OVERVIEW OF PCML

Learning an optimal personalized model $f^i$ for the $i$-th client expects a sufficient utilization of the data, sharing a common distribution from other clients. However, due to privacy concerns, one cannot identify these overlaps with direct access to the raw data. We suggest leveraging the overlaps via the learned data manifold to prevent privacy leakage. As shown in Figure 4, in general, our proposed precisely collaborative learning scheme contains:

- for the data in other clients, we aim to precisely learn from the shared overlaps identified by the local data manifold, as shown in Figure 4 (a) (b) (c);
- for the data generated from $\mathcal{M}^i$, we expect to advance models with the generated data in an optimal sampling density $p_z^i(z)$, as shown in Figure 4 (d).

### 4.3 PRECISION COLLABORATION I: LEARNING FROM THE SHARED OVERLAPS

From Figure 1 (a), different clients could share different distribution overlaps, and the distribution overlaps are associated with the overlapped region of the local data manifolds. While the data manifold of local clients is mostly unknown and can hardly be inferred from limited local data, we propose to learn the global data manifold with the data on all clients. In this way, all data are utilized and contribute to manifold learning.

**Learn the global manifold.** The data $z \in D$ are usually supported on an unknown lower-dimensional manifold $\mathcal{M}$. In our realization, we propose to use an NF method to learn the global manifold $\mathcal{M}^g$, and the learning of the NF is shown in Figure 3. In particular, we learn two bijective transformations $g_\theta$ and $h_\phi$ to transform the data $z$ to the latent variable $u'$, i.e.,

$$u' = h_\phi \circ \text{Split} \circ g_\theta(z), \tag{6}$$

where $u'$ is the learned representation of the sample $z$. Please note that in the rest of this paper we will use $g_\theta^*$ to denote $\text{Split} \circ g_\theta$ and $g_\theta^{*-1}$ denotes $g_\theta^{-1} \circ \text{Pad}$.

With the **Split** operation, we learn a diffeomorphism from the data $z \sim p_z^g(z)$ to a lower dimension space $u' \sim p_{u'}(u')$ as shown in Figure 4 (a). This means that we transform the original data manifold

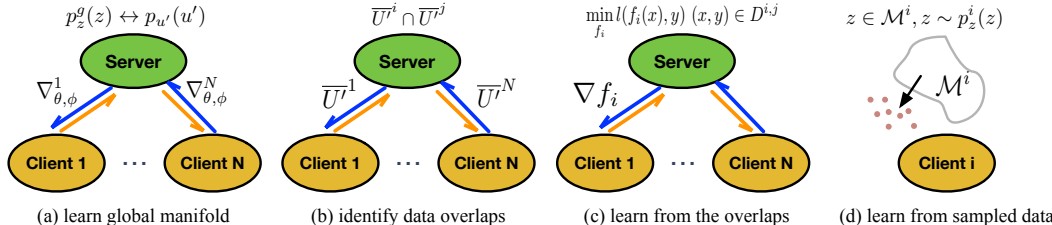

Figure 4: (a) all clients collaboratively learn the global manifold, which transforms $p_z^g$ to a given density $p_{u'}(u')$; (b) identify the data overlaps by the determined projected manifold $\overline{U'}^i$; (c) the clients train their models using the identified data overlaps ($D^{i,j}$) from others; (d) local clients train their models using the sampled data with the learned density $p_z^i(z)$.

$\mathcal{M}^g$ to the projected data manifold $U'$. Note that the decoder is the inverse of the encoder. The data is reconstructed given the latent variable $u' \in U'$, i.e., $z = g_\theta^{*-1} \circ h_\phi^{-1}(u')$.

**Determine the local manifold.** From Figure 1, the global manifold $\mathcal{M}^g$ contains the local manifold $\mathcal{M}^i$. A local data manifold $\mathcal{M}^i$ should contain the local data $D^i$. Considering that the original global manifold $\mathcal{M}^g$ and the latent space $U'$ ($U' = \mathbb{R}^{d'}$) are topologically equivalent, we propose to approximate the local manifold with the learned representations:

$$\mathcal{M}^i = g_\theta^{*-1} \circ h_\phi^{-1}(\overline{U'^i}), \ where \ U'^i = \left\{ h_\phi \circ g_\theta^*(x_j^i) \right\}_{j=1}^{n^i}, \tag{7}$$

where $U'^i$ denotes the set of the samples transformed to $U'$ from $D^i$, and $\overline{U'^i}$ is called the projected local data manifold. It is computed as the convex hull of $U'^i$, which continuously contains all representations. For example, if $U'^i$ contains two representations $\{[0.1, 0.2], [0.2, 0.1]\}$, the convex hull $\overline{U'}^i$ is a line segment in $\mathbb{R}^2$, which is from the coordinate point $[0.1, 0.2]$ to $[0.2, 0.1]$[1].

**Identify the data overlaps from other clients.** Since we cannot determine the data overlaps directly because of privacy concerns, we propose to identify the overlaps using the learned projected manifolds $\overline{U'}^i$ as shown in Figure 4 (b). Note that the data overlaps correspond to the overlaps of the data manifolds. For example, suppose $D^{i,j} \subset D^j$ is a subset of $D^j$, if $D^{i,j}$ lies on $\mathcal{M}^i$, $D^{i,j}$ is the data overlap between $D^i$ and $D^j$. From Eq.(7), $\mathcal{M}^i$ is reconstructed by $g_\theta^{*-1} \circ h_\phi^{-1}$ with $\overline{U'^i}$. Therefore, $D^{i,j}$ could be identified as follows:

$$D^{i,j} = \left\{ z_k^j | h_\phi \circ g_\theta^*(z_k^j) \in \overline{U'^i}, \forall \, z_k^j \in D^j \right\} \subset \mathcal{M}^i. \tag{8}$$

From Eq.(8), $D^{i,j}$ is the data in $D^j$ whose representation $h_\phi \circ g_\theta^*(z_k^j)$ fall into the projected manifold of $D^i$.

By learning from the overlaps identified from other clients, we have the following objective to train $f_i$ for the $i$-th client shown in Figure 4 (c),

$$\min_{f_i \in \mathcal{F}} \frac{1}{N-1} \sum_{k=0, k \neq i}^{N-1} \mathbb{E}_{(x^k, y^k) \in D^{i,k}}(\ell(f_i(x^k), y^k)). \tag{9}$$

## 4.4 PRECISION COLLABORATION II: LEARNING WITH AN OPTIMAL SAMPLING DENSITY

In Sec. 4.3, we learn personalized models from the data overlaps between clients. However, the model performance on the unshared data cannot be improved by collaborating with others. The specific region $\mathcal{M}_s^i$ has no overlap with others, which is formulated as

$$\mathcal{M}_s^i = g_\theta^{*-1} \circ h_\phi^{-1}(\overline{U'^i_s}), \ where \ \overline{U'^i_s} = \overline{U'^i} - \cup_{j=0, j \neq i}^{N-1}(\overline{U'^j} \cap \overline{U'^i}), \tag{10}$$

---

[1]The computation of the convex hull of a high-dimensional point set may be time-consuming. We approximate the convex hull, and the computation method is in Appendix.

We propose to advance the model by generating data sampled from the local manifold $\mathcal{M}^i$.

While an arbitrary sampling density could generate data $D'^i$ deviated from the local distribution $d(D'^i, D^i) > \epsilon$, this could induce bias to the learned model. An optimal utilization of the synthetic data expects a sampling density close to $p_z^i(z)$. Therefore, we propose to sample from $\mathcal{M}^i$ with the exact estimation of $p_z^i(z)$.

**Exact likelihood estimation.** Note that we learn the manifold by applying a normalizing flow framework, which achieves the exact likelihood estimation simultaneously.

Since we learn the global data manifold, the global data density $p_z^g(z)$ is transformed to $p_{u'}(u')$. For the local data density $p_z^i(z)$, we have the following proposition.

**Proposition 1.** *(proof in Appendix) For any data point $z \in \mathcal{M}_s^i$, the local density $p_z^i(z)$ satisfies*

$$
\begin{aligned}
p_z^i(z) =& c \cdot p_{u'}\left(h_\phi \circ g_\theta^*(z)\right) \left|\det J_{h_\phi}\left(h_\phi \circ g_\theta^*(z)\right)\right|^{-1} \\
& \left|\det \left[J_{g_\theta^*}^T\left(g_\theta^*(z)\right) J_{g_\theta^*}\left(g_\theta^*(z)\right)\right]\right|^{-\frac{1}{2}},
\end{aligned}
\tag{11}
$$

*where $c$ is a proportionality constant, and $J_{h_\phi}$ and $J_{g_\theta^*}$ are the Jacobian matrix of $h_\phi$ and $g_\theta^*$, respectively.*

From the Proposition 1, to sample $z \in \mathcal{M}_s^i$ in the density $p_z^i(z)$, we could firstly sample $u' \sim p_{u'}(u')$ and choose $u' \in \overline{U_s'^i}$ defined in Eq.(10). Then we transform the sampled $u'$ to the data space by $z = g_\theta^{*-1} \circ h_\phi^{-1}(u')$ as shown in Figure 4 (d). The objective for the generated data is formulated as follows

$$
\min_{f_i \in \mathcal{F}} \mathbb{E}_{(x,y) \sim p_z^i(z)} \ell(f_i(x), y),
\tag{12}
$$

where the sampled $(x, y)$ in the third term satisfies $(x, y) \in \mathcal{M}_s^i$.

Combining the identified data overlaps, the final objective could be formulated as follows

$$
\begin{aligned}
\min_{f_i \in \mathcal{F}} & \frac{1}{n^i} \sum_{j=1}^{n^i} \ell(f_i(x_j^i), y_j^i) \\
& + \alpha \cdot \frac{1}{N-1} \sum_{k=0, k \neq i}^{N-1} \mathbb{E}_{(x^k, y^k) \in D^{i,k}}\left(\ell(f_i(x^k), y^k)\right) \\
& + \beta \cdot \mathbb{E}_{(x,y) \sim p_z^i(z)} \ell(f_i(x), y),
\end{aligned}
\tag{13}
$$

where $\alpha, \beta > 0$ are the regularization parameters.

## 4.5 THE REALIZATION OF PCML

**Learning the global manifold $\mathcal{M}^g$ and identifying overlaps.** Following the work in (Brehmer & Cranmer, 2020), we train $g_\theta$ and $h_\phi$ by a two-stage optimization framework. In particular, we first train $g_\theta$ to obtain the projection onto the manifold by minimizing the reconstruction error. Then, we optimize $h_\phi$ to approximate the density by maximizing the likelihood (Brehmer & Cranmer, 2020).

After learning the $\mathcal{M}^g$, the local manifold $\mathcal{M}^i$ and the overlaps among clients could be identified according to Eq.(7) and Eq.(8).

**Sampling data with the learned local manifold $\mathcal{M}^i$.** We approximate the exact likelyhood in Eq.(11) when learning the NF model. The data could be generated from the determined $\mathcal{M}_s^i$ in Eq.(10). More implementation details could be found in the Appendix.

## 4.6 DISCUSSION ON PRIVACY OF PCML

In the realization of PCML, we try our best to achieve privacy protection as other distributed learning scenario methods or FL methods. In particular,

**1. the model training :** during the training of the manifold $\mathcal{M}^g$ and the local model $f_i$, only the sum of the gradients is allowed to be transferred to the server. The raw data and the invertible representations are not allowed to be shared thoroughly.

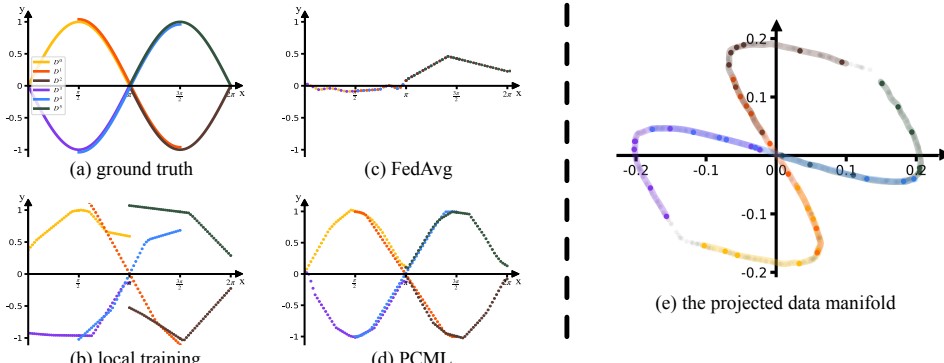

Figure 5: Illustrations of the synthetic experiments. (a) The learning tasks of the six clients; (b), (c), and (d) are the performance of the models learned by local training, FedAvg and PCML; (e) the learned projected global data manifolds. The points denote the samples from different clients. The colored lines denote the identified local manifolds.

**2. the computation of the overlaps:** the server computes the overlaps between clients using the projected manifold $\overline{U'}^i$ uploaded by local clients, and $\overline{U'}^i$ has no information about the raw data or the invertible representations.

## 5 EXPERIMENTS

### 5.1 SYNTHETIC EXPERIMENTS

Firstly, we show the motivation of our method by conducting experiments on synthetic data [2].

**Synthetic data.** Suppose there are 96 clients: $D^i, i \in \{1, 2..., 96\}$. The data points $z = \{x, y\}$ is generated from two objectives $y = sin(x) + \epsilon$ or $y = -sin(x) + \epsilon$ shown in Figure 5 (a), where $\epsilon \sim \mathcal{N}(0, 0.1)$ denotes label noise.

**Fragmented data overlaps.** To generate heterogeneous and overlapped local data, we sample $x$ from the overlapped ranges. In particular, we separate the input space $X$ into four intervals $[0, \frac{\pi}{2}]$, $[\frac{\pi}{2}, \pi]$, $[\pi, \frac{3\pi}{2}]$ and $[\frac{3\pi}{2}, 2\pi]$, and each client randomly chooses two different intervals to sample data. To create conflicting learning tasks, the label of the selected 48 clients is calculated by $y = sin(x) + \epsilon$, and the label of the remaining 48 clients is calculated by $y = -sin(x) + \epsilon$.

In this setting, learning a global model for all clients could hurt the model performance as there are two conflicting learning tasks shown in Figure 5 (c). The best way of collaborative model learning is identifying the data overlaps, which are sampled from the identical objective with the same intervals. For example, $D^0$ consists of the data sampled from $[0, \frac{\pi}{2}]$ and $[\frac{\pi}{2}, \pi]$, while $D^1$ consists of the data sampled from $[\frac{\pi}{2}, \pi]$ and $[\pi, \frac{3\pi}{2}]$ shown in Figure 5 (a). Learning an optimal model for $D^0$ needs to precisely identify the data overlap sampled from $[\frac{\pi}{2}, \pi]$ in $D^1$. From Figure 5 (e), PCML efficiently obtains local data manifolds and identifies the data overlaps between clients. Therefore, PCML learns a better model by precision collaboration, which maximizes the benefits and avoids potential negative transfer from other clients as shown in Figure 5 (d).

### 5.2 BENCHMARK EXPERIMENTS

**Datasets.** We adopt three benchmark image datasets: CIFAR10 (Krizhevsky et al., 2009), FEM-NIST (Caldas et al., 2018) and a tabular dataset *Adult* (Kohavi et al., 1996). We create the federated environment with data heterogeneity for CIFAR10 by randomly allocating several classes to each client following the work (McMahan et al., 2017). We use $K$ to denote the number of clients and $S$ to denote the number of classes in each client. For CIFAR10, $K = 150, S = 5$ means there are 150 clients and each client contains 5 classes of images. For FEMNIST which has 10 classes of

---

[2]The source codes are made publicly available at `https://anonymous.4open.science/r/PCML-0EBA`.

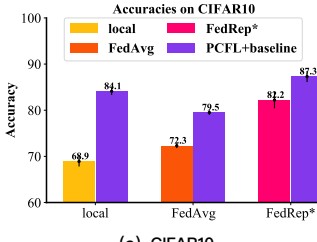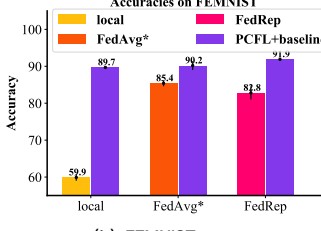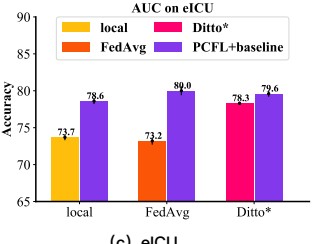

Figure 6: Visualization of the experimental results on CIFAR10, FEMNIST and eICU.

handwritten letters, we consider the setting of $K = 200, S = 5$. The number of samples in each client is determined according to a log-normal distribution (Li et al., 2019a). For the tabular dataset *Adult*, the task is to predict whether an individual's income is beyond 50K/year based on some census features, including age, race, workclass, etc. Following the setting in (Mohri et al., 2019), all individuals are split into two clients. One is PhD client and the other is non-PhD client.

**Baselines.** We compare our method with various baselines, including global and personalized methods. Global baselines include: 1) FedAvg (McMahan et al., 2017); 2) FedProx (Li et al., 2020); 3) SCAFFOLD-M (Cheng et al., 2024). Personalized baselines include: 1) Fed-MTL (Smith et al., 2017); 2) PerFedAvg (Fallah et al., 2020); 3) LG-FedAvg (Liang et al., 2020); 4) FedPer (Arivazhagan et al., 2019); 5) FedRep (Collins et al., 2021); 6) APFL (Deng et al., 2020); 7) L2GD (Hanzely & Richtárik, 2020); 8) Ditto (Li et al., 2021b); 9) kNN-Per (Marfoq et al., 2022).

**Experimental Results.** The accuracy of all methods on CIFAR10 dataset are shown in Table 1 and Figure 6.PCML outperforms all baselines on this classification task. Since each client has insufficient data samples ($n^i = 333$), FedAvg (72.3%) learning from all data has a better performance compared with local (68.9). FedRep (82.2%) surpasses other baselines by learning a global feature extractor. As a pluggable method, PCML could be used to enhance the performance of other art methods. From Figure 6, PCML improves the performance of FedRep by 5.1%, which indicates that PCML effectively identifies the informative knowledge from others.

Table 1: Experiments on CIFAR10, FEMNIST, and eICU (%).

| Methods | CIFAR10 (ACC) | FEMNIST (ACC) | eICU (AUC) |
|---|---|---|---|
| local | $68.9_{\pm 1.1}$ | $59.9_{\pm .9}$ | $73.7_{\pm 1.4}$ |
| FedAvg | $72.3_{\pm .5}$ | $85.4_{\pm .8}$ | $73.2_{\pm .5}$ |
| FedProx | $71.5_{\pm .8}$ | $84.9_{\pm 1.7}$ | $78.2_{\pm .2}$ |
| SCAFFOLD-M | $72.8_{\pm 1.3}$ | $85.3_{\pm .9}$ | $70.5_{\pm .3}$ |
| Fed-MTL | $68.4_{\pm 2.2}$ | $60.7_{\pm 2.2}$ | $77.2_{\pm 1.6}$ |
| PerFedAvg | $67.3_{\pm .1}$ | $26.8_{\pm .5}$ | $73.8_{\pm .3}$ |
| LG-FedAvg | $69.2_{\pm .3}$ | $35.2_{\pm 1.4}$ | $74.5_{\pm .2}$ |
| FedPer | $82.2_{\pm .9}$ | $82.9_{\pm .2}$ | $74.3_{\pm .7}$ |
| FedRep | $82.2_{\pm 1.8}$ | $82.8_{\pm 1.4}$ | $74.1_{\pm 1.2}$ |
| APFL | $64.5_{\pm 3.7}$ | $62.3_{\pm 1.4}$ | $68.3_{\pm .8}$ |
| L2GD | $10.0_{\pm .0}$ | $14.5_{\pm 1.6}$ | $72.0_{\pm .6}$ |
| Ditto | $82.1_{\pm .2}$ | $85.4_{\pm .2}$ | $78.3_{\pm .1}$ |
| kNN-Per | $79.4_{\pm .4}$ | $86.1_{\pm .8}$ | $75.2_{\pm .3}$ |
| PCML | $\mathbf{87.3}_{\pm 1.2}$ | $\mathbf{91.9}_{\pm .4}$ | $\mathbf{80.0}_{\pm .6}$ |

## 6 CONCLUSION

In this paper, we propose a precise collaboration framework PCML for a more general FL learning scenario, where the fragmented and shared knowledge is distributed among other clients. Experiments on benchmark datasets and a real-world clinical dataset verify the superiority of our method because of optimal and precise utilization of the shared information. Our framework determines the overlaps between clients, which suggests several attractive topics, such as identifying malicious clients or noisy data, etc.

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

## A THEORETICAL PROOFS

### A.1 PROOF OF PROPOSITION 1

Suppose there is a smooth and injective mapping $g_\theta^* : \mathbb{R}^d \to \mathbb{R}^{d'}$ with $d' \leq d$, $U \in \mathbb{R}^{d'}$ is the latent variable and has $Z := g_\theta^{*-1}(U)$. From differential geometry (Krantz & Parks, 2008), we have

$$p_z(z) = p_u\left(g_\theta^*(z)\right) \left| \det J_{g_\theta^*}^\top \left(g_\theta^*(z)\right) J_{g_\theta^*}\left(g_\theta^*(x)\right) \right|^{-1/2}. \tag{14}$$

Suppose there is a smooth and bijective mapping $h_\phi : \mathbb{R}^{d'} \to \mathbb{R}d'$, $U' \in \mathbb{R}^{'}$ is the latent variable which has $U := h_\phi(U')$. We have

$$p_u(u) = p_{u'}\left(h_\phi(u)\right) \left| \det J_{h_\phi}(h_\phi(u)) \right|^{-1}. \tag{15}$$

According to the chain rule, combining Eq.(14) and Eq.(15), we have

$$p_z(z) = p_{u'}\left(h_\phi \circ g_\theta^*(z)\right) \left| \det J_{h_\phi}(h_\phi \circ g_\theta^*(z)) \right|^{-1} \left| \det J_{g_\theta^*}^\top \left(g_\theta^*(z)\right) J_{g_\theta^*}\left(g_\theta^*(z)\right) \right|^{-1/2}. \tag{16}$$

Since we learn a global manifold $\mathcal{M}^g$ with the data from all clients, the density of the data from all clients $p_z^g(z)$ is approximated in Eq.(16). From the definition of $\mathcal{M}_s^i$ in Eq.(10), if there is a data point $z \in \mathcal{M}_s^i$, $z$ will cannot be sampled from any other manifolds $\mathcal{M}^j$ ( $j \neq i$ ) but $\mathcal{M}^i$, i.e.,

$$\forall z \in \mathcal{M}^g, \quad s.t., \quad z \notin \mathcal{M}^j (j \neq i) \; if \; z \in \mathcal{M}_s^i. \tag{17}$$

Therefore, we have

$$p_z^g(z, z \in \mathcal{M}_s^i | z \in \mathcal{M}^i) = \frac{p_z^g(z, z \in \mathcal{M}_s^i, z \in \mathcal{M}^i)}{p_z^g(z \in \mathcal{M}^i)} = \frac{p_z^g(z, z \in \mathcal{M}_s^i)}{p_z^g(z \in \mathcal{M}^i)} = \frac{1}{p_z^g(z \in \mathcal{M}^i)} \cdot p_z^i(z, z \in \mathcal{M}_s^i). \tag{18}$$

Combining with Eq.(16), for $z \in \mathcal{M}_s^i$, we have

$$p_z^i(z) = c \cdot p_{u'}(h_\phi \circ g_\theta^*(z)) \left| \det J_{h_\phi}(h_\phi \circ g_\theta^*(z)) \right|^{-1} \left| \det J_{g_\theta^*}^\top (g_\theta^*(z)) J_{g_\theta^*}(g_\theta^*(z)) \right|^{-1/2}, \tag{19}$$

and Proposition 1 holds.

---

**Algorithm 1** Learn the global manifold in the federated learning setting

---

**Input:** epoch $T_m$, batch size $B_m$, initial manifold model $\mathcal{M}^g$ with the parameters $\theta$ and $\phi$.

1: **for** $t = 0, ..., T_m - 1$ **do**
2:      randomly select a subset of clients $S_t$
3:      **for** client $D^i \in S_t$ in parallel **do**
4:          draw mini-batch $z^i : z_{t_1}^i, ..., z_{t_{B_m}}^i \sim D^i$
5:          **if** $t < T_m/2$ **then**
6:              calculate the loss: $\frac{1}{B_m} \sum_{i=1}^{B_m} \| z^i - g_\theta^{-1}(g_\theta(z^i)) \|$;
7:              then calculate the gradients of loss with respect to parameters $\theta$;
8:          **else**
9:              calculate the loss: $-\frac{1}{B_m} \sum_{i=1}^{B_m} \left( \log p_{u'}(h_\phi \circ g_\theta^*(z^i)) - \log \det J_h(h_\phi \circ g_\theta^*(z^i)) \right)$;
10:             then calculate the gradients of loss with respect to parameters $\phi$;
11:          **end if**
12:      **end for**
13:      **Server** aggregates the gradients of selected clients and update the parameters $\theta$ and $\phi$.
14: **end for**
15: **Output:** the learned manifold model $g_\theta$ and $h_\phi$.

---

## B  PIPELINE OF OUR FRAMEWORK PCML

The pipeline of the global manifold learning $\mathcal{M}^g$ is elaborated in Algorithm 1. We learn a global manifold model in the federated learning setting. There are two phases of training. Firstly, only the parameters of $g_\theta$ are updated as in Line 5-7. Then the parameters of $h_\phi$ are updated as in Line 9-10. The learned manifold model $g_\theta^* \circ h_\phi$ is utilized in our framework PCML, whose pipeline is elaborated in Algorithm 2. To begin with, the local manifolds of clients are extracted based on Eq.(7) and the distribution overlaps are calculated based on Eq.(8). Since only the borders of convex hulls are exchanged, there is no leakage of sensitive information. The data from the overlapped distribution of other clients are used to train the models. They are utilized by transmitting the average gradients through the server as in Line 7-14.

## C  MORE DISCUSSIONS ABOUT PCML

### C.1  A NEW METRIC OF CLIENT SIMILARITY

Our framework PCML inspires a novel metric for measuring the similarity between local clients. For example, suppose the $i$-th client and $j$-th client has the identical local manifold $\mathcal{M}^i = \mathcal{M}^j$, the similarity between clients is close to 1. On the contrary, if the two local manifolds are disjoint $\mathcal{M}^i \cap \mathcal{M}^j = \emptyset$, the measured similarity should be 0. In particular, we propose to measure the similarity as the Intersection of Union (IoU) of the projected local manifold,

$$S(D^i, D^j) = \text{IoU}(\overline{U'^i}, \overline{U'^j}). \tag{20}$$

---

**Algorithm 2** The learning framework `PCML`

---

**Input:** epoch $T$, batch size $B$, initial models $\{f^0, ..., f^{N-1}\}$, hyperparameters $\alpha$ and $\beta$;

1: all the clients determine the local manifold $\mathcal{M}^i$ and $\overline{U'^i}$ based on Eq.(7), and send $\overline{U'^i}$ to the **Server**.
2: the **Server** calculates the overlaps of $\overline{U'^i}$ between clients, calculates $\overline{U'^i_s}$ based on Eq.(10), and sends them to each client;
3: **for** $t = 0, ..., T-1$ **do**
4:    randomly select a subset of clients $S_t$,
5:    the selected clients send their local models to the **Server**;
6:    **for** client $D^i \in S_t$ in parallel **do**
7:       draw mini-batch $(x^i, y^i) \sim D^i$;
8:       calculate the loss $\mathbb{E}_{(x^i,y^i)\in D^i}(\ell(f_i(x^k), y^k)) + \beta \cdot \mathbb{E}_{(x,y)\in p^i_z(z)}\ell(f_i(x), y)$, and update the model $f^i$ using the gradients of loss;
9:       **for** $k = 0, ..., N, k \neq i$ **do**
10:          draw mini-batch $(x^k, y^k) \sim D^{i,k}$
11:          calculate the loss $\alpha \cdot \mathbb{E}_{(x^k,y^k)\in D^{i,k}}\ell(f_i(x^k), y^k)$, and update the model $f^i$ using the gradients of loss;
12:       **end for**
13:       the **Server** aggregates the parameters of $f^i$ from other clients and send the average to the $i$-th client;
14:       then the $i$-th client $D^i$ updates the model $f^i$ with the received parameters and local gradients.
15:    **end for**
16: **end for**
17: **Output:** the learned personalized models $\{f^0, ..., f^{N-1}\}$.

---

**A communication-efficient client-level collaboration.** Our proposed metric allows efficient collaborator identification which reduces the communication and computation overhead. For example, we require $D^i$ to collaborate with certain clients who have a higher client similarity:

$$\min_{f\in\mathcal{F}} \frac{1}{\sum_{k=0, S(D^i, D^k)\geq\epsilon}^{N-1} n^k} \sum_{k=0, S(D^i, D^k)\geq\epsilon}^{N-1} \sum_{i=1}^{n^k} l\left(f\left(x_i^k\right), y_i^k\right), \tag{21}$$

where $\epsilon \geq 0$ is a pre-defined threshold. Note that the objective in Eq.(21) is different from clustered FL methods. Clustered FL methods learn a common model for each cluster while Eq.(21) learns a personalized model for each client. Experimental results shown in Sec. D.1 verify that this method achieves a comparable performance while reducing communication and computation overhead.

Previous work has explored the problem of identifying similar datasets in a graph network for downstream learning tasks (Hallac et al., 2015). In particular, Jung (2020) formulate the learning from distributed local datasets as a convex optimization problem and proposes to cluster the local datasets according to the learned parameters. Jung & Tran (2019) extend network lasso methods in regression tasks under a clustering assumption. These cluster-based methods could be applied in federated learning with a proper design for privacy-preserving. In our experiments, we use network lasso to cluster the local datasets under the federated setting. In Table 2, we show the comparison of `PCML` and the clustered methods. Our method outperforms all cluster-based methods, which demonstrates that a precision identification of overlaps in other clients facilitates model learning. Moreover, an interesting direction is the application of our proposed similarity metric in the graph network. For example, the manifold learning of local datasets in the graph network may also be used for similarity measurement.

## C.2    EVALUATION ABOUT THE PROPOSED NOVEL METRIC

In Sec. C.1, we propose a novel metric for measuring the distance between local clients, which could be used for a communication-efficient client-level collaboration. We conduct experiments on eICU dataset, in which we select the most similar 7 clients for each client to learn a personalized model. The experimental results are shown in Table 3.

Table 2: More experimental results on eICU

| Methods | AUC (%) |
|---|---|
| local | $73.7_{\pm1.4}$ |
| FedAvg | $73.2_{\pm.5}$ |
| FedProx | $78.2_{\pm.2}$ |
| Fed-MTL | $77.2_{\pm1.6}$ |
| PerFedAvg | $73.8_{\pm.3}$ |
| LG-FedAvg | $74.5_{\pm.2}$ |
| FedPer | $74.3_{\pm.7}$ |
| FedRep | $74.1_{\pm1.2}$ |
| APFL | $68.3_{\pm.8}$ |
| L2GD | $72.0_{\pm.6}$ |
| Ditto | $78.3_{\pm.1}$ |
| Clustered FL | $74.7_{\pm.3}$ |
| Network Lasso | $76.3_{\pm.8}$ |
| ours | $80.0_{\pm.6}$ |

From Table 3, our method for identifying the collaborators achieves a comparable performance compared with baselines and reduces computation and communication overhead by collaborating with a subset of local clients.

To explore the effect of $\epsilon$ on the performance of the learned models, we set $\epsilon$ by controlling the number of clients to collaborate for each client. There are 14 clients in eICU dataset. We test the number of the collaborator ($C$) to be 1, 3, 5, ... etc. The results are shown in Table 4. When $C = 7$, the learned model achieves the highest AUC (78.0). When $C > 7$, the performance tends to remain unchanged.

Table 3: Experimental results on eICU

| Methods | AUC (%) |
|---|---|
| local | $73.7_{\pm1.4}$ |
| FedAvg | $73.2_{\pm.5}$ |
| FedProx | $78.2_{\pm.2}$ |
| Fed-MTL | $77.2_{\pm1.6}$ |
| PerFedAvg | $73.8_{\pm.3}$ |
| LG-FedAvg | $74.5_{\pm.2}$ |
| FedPer | $74.3_{\pm.7}$ |
| FedRep | $74.1_{\pm1.2}$ |
| APFL | $68.3_{\pm.8}$ |
| L2GD | $72.0_{\pm.6}$ |
| Ditto | $78.3_{\pm.1}$ |
| ours | $78.0_{\pm.1}$ |

Table 4: Experimental results on eICU with adaptive $\epsilon$

| $C$ | AUC (%) |
|---|---|
| 1 | $69.0_{\pm.4}$ |
| 3 | $75.4_{\pm.2}$ |
| 5 | $75.5_{\pm.7}$ |
| 7 | $78.0_{\pm.1}$ |
| 9 | $77.1_{\pm.9}$ |
| 11 | $76.8_{\pm.2}$ |
| 13 | $77.0_{\pm.3}$ |

## C.3 COMPUTATION COMPLEXITY AND OPTIMIZATION EFFICIENCY

From Algorithm 1 and Algorithm 2, PCML is realized by a two-staged optimization framework. For the training of the normalizing flow in the first stage, PCML learns a global model for all clients, which has the computation complexity as FedAvg. For the identification of the manifold overlaps,

it has $O(1)$ time complexity as the server only computes it once. For the training of local models in the second stage, PCML learns a personalized model for each client, which has the computation complexity as other personalized methods. From the above all, PCML achieves a similar computation complexity as baselines.

A classical NF method requires a fixed dimensionality of the latent space, which is the same as the dimension of the data. In this case, learning such a NF model could bring a huge computation overhead when the data is high-dimensional. In the first phase of our framework, we learn a low-dimensional manifold in a NF method, which significantly reduces the computation overhead.

Our method learns from local data, data overlaps of other clients, and sampled data in the manifold. By precision collaboration, we avoid learning from all data. We make comparisons of run-time consumption with the baselines. The experiments are conducted on the same device NVIDIA GeForce RTX 2080 Ti. The results on eICU dataset are displayed in Table 5. As a pluggable method, the time consumption of PCML is comparable to the corresponding baselines. Fed-MTL involves computing the correlation of the parameters among all client models, which could result in more computation overhead.

Table 5: Run-time consumption comparisons

| Methods | Run-time consumption |
|---------|---------------------|
| local | 33 min 47 s |
| FedAvg | 57 min 41 s |
| FedProx | 56 min 35 s |
| Fed-MTL | 101 min 12 s |
| PerFedAvg | 79 min 47 s |
| LG-FedAvg | 55 min 27 s |
| FedPer | 57 min 43 s |
| FedRep | 40 min 37 s |
| APFL | 92 min 13 s |
| L2GD | 63 min 4 s |
| Ditto | 71 min 34 s |
| PCML | 74 min 27 s |

### C.4 PRIVACY PRESERVING

PCML maintains data confidentiality as baselines because there is no shared data between local clients. PCML achieves privacy-preserving as baselines because our framework learns models by communicating model parameters only. Federated learning may need further exploration to maintain data privacy. Some researchers claim there is information leakage when sharing models or gradients (Zhu et al., 2019). To alleviate this issue, there are research proposing to apply other techniques to FL methods, such as differential privacy (Wei et al., 2020), secure multi-party computation, etc. PCML is also compatible with these techniques.

## D ABLATION STUDIES

### D.1 MORE ABLATION STUDIES ABOUT PCML

PCML is pluggable for other algorithms. We test local, FedAvg, FedRep and Ditto which are implemented with/without our method as in Table 6 and Table 7. In the dataset Adult, all individuals are split into two clients, one of which is PhD client and the other is non-PhD client. The non-PhD client contains 32148 training samples while the PhD client contains 413 samples. Therefore the non-PhD client of Adult can not benefit much from other methods. In other datasets, our method boosts the baselines by large margins.

Table 6: Experiment results of PCML implemented on CIFAR10, FEMNIST, and CelebA (%)

| Dataset | local | PCML (local) | FedAvg | PCML (FedAvg) | FedRep | PCML (FedRep) |
|---------|-------|--------------|--------|---------------|--------|---------------|
| CIFAR10 | $68.9_{\pm 1.1}$ | $84.1_{\pm .8}$ ($\uparrow$ 15.2) | $72.3_{\pm .5}$ | $79.5_{\pm .5}$ ($\uparrow$ 7.2) | $82.2_{\pm 1.8}$ | $87.3_{\pm 1.2}$ ($\uparrow$ 5.1) |
| FEMNIST | $59.9_{\pm .9}$ | $89.7_{\pm .2}$ ($\uparrow$ 29.8) | $85.4_{\pm .8}$ | $90.2_{\pm 1.2}$ ($\uparrow$ 4.8) | $82.8_{\pm 1.4}$ | $91.9_{\pm .4}$ ($\uparrow$ 9.1) |
| CelebA | $69.3_{\pm 1.1}$ | $85.8_{\pm 1.1}$ ($\uparrow$ 16.5) | $85.2_{\pm 2.1}$ | $89.5_{\pm 2.0}$ ($\uparrow$ 4.3) | $68.1_{\pm .6}$ | $71.2_{\pm .6}$ ($\uparrow$ 3.1) |

Table 7: Experiment results of PCML on eICU and Adult (%)

| Dataset | local | PCML (local) | FedAvg | PCML (FedAvg) | Ditto | PCML (Ditto) |
|---|---|---|---|---|---|---|
| Adult non-PhD | $83.4_{\pm.1}$ | $83.6_{\pm.1}$ ($\uparrow$ .2) | $83.4_{\pm.3}$ | $83.6_{\pm.1}$ ($\uparrow$ .2) | $83.5_{\pm.2}$ | $83.6_{\pm.0}$ ($\uparrow$ .1) |
| Adult PhD | $70.2_{\pm.4}$ | $77.3_{\pm.7}$ ($\uparrow$ 7.1) | $72.9_{\pm.2}$ | $76.8_{\pm.2}$ ($\uparrow$ 3.9) | $75.7_{\pm.9}$ | $76.8_{\pm.1}$ ($\uparrow$ 1.1) |
| eICU | $73.7_{\pm1.4}$ | $78.6_{\pm.4}$ ($\uparrow$ 4.9) | $73.2_{\pm.5}$ | $80.0_{\pm.6}$ ($\uparrow$ 6.8) | $78.3_{\pm.1}$ | $79.6_{\pm.4}$ ($\uparrow$ 1.3) |

# E  EXPERIMENTS AND IMPLEMENTATION DETAILS

## E.1  EXPERIMENTS ON MORE HETEROGENEOUS SETTINGS

We also use CelebA dataset to verify the effectiveness of our method. The task on CelebA is to classify whether the celebrity in the image is smiling (Li et al., 2021b). There are 545 clients and 21 samples per client in average.

The experimental results on CelebA are shown in Table 9 and Figure 7. Moreover, we conduct experiments on FEMNIST on more heterogeneous settings with more clients. We partition the dataset into 400 clients with the Dirichlet distribution $Dir_{400}(0.1)$ and $Dir_{400}(0.5)$ following the work in (Wang et al., 2019). We compare our method with the baselines, and the results are shown in Table 8. With more clients, each client has fewer training samples. Local method shows poor performance (63.6% in $Dir_{400}(0.5)$). Global methods (FedAvg and FedProx) achieve better performance under a less heterogeneous setting ($Dir_{400}(0.5)$), while the performance of personalized methods degrades. Under two settings ($Dir_{400}(0.1)$ and $Dir_{400}(0.5)$), PCML outperforms all baselines by identifying the informative overlaps for each client.

Table 8: More experimental results on FEMNIST (Acc %)

| methods | $Dir_{400}(0.1)$ | $Dir_{400}(0.5)$ |
|---|---|---|
| local | $71.8_{\pm.8}$ | $63.6_{\pm.4}$ |
| FedAvg | $69.1_{\pm.5}$ | $80.6_{\pm1.7}$ |
| FedProx | $67.8_{\pm.3}$ | $79.9_{\pm.2}$ |
| Fed-MTL | $81.8_{\pm.9}$ | $60.1_{\pm.4}$ |
| PerFedAvg | $82.4_{\pm1.1}$ | $46.7_{\pm.8}$ |
| LG-FedAvg | $86.8_{\pm.8}$ | $49.5_{\pm.5}$ |
| FedPer | $91.1_{\pm.3}$ | $76.8_{\pm.2}$ |
| FedRep | $91.8_{\pm1.4}$ | $74.6_{\pm.3}$ |
| APFL | $79.9_{\pm.9}$ | $60.9_{\pm.5}$ |
| L2GD | $77.6_{\pm.4}$ | $39.9_{\pm1.5}$ |
| Ditto | $91.7_{\pm.7}$ | $82.9_{\pm.4}$ |
| PCML | $96.1_{\pm.4}$ | $88.3_{\pm.6}$ |

Table 9: CeleA (Acc %)

| Methods | ACC (%) |
|---|---|
| local | $69.3_{\pm1.1}$ |
| FedAvg | $85.2_{\pm2.1}$ |
| FedProx | $81.2_{\pm1.2}$ |
| Fed-MTL | $68.2_{\pm.4}$ |
| PerFedAvg | $68.6_{\pm.8}$ |
| LG-FedAvg | $68.4_{\pm1.2}$ |
| FedPer | $68.2_{\pm.5}$ |
| FedRep | $68.1_{\pm.6}$ |
| APFL | $71.4_{\pm.6}$ |
| L2GD | $67.9_{\pm2.0}$ |
| Ditto | $84.5_{\pm.6}$ |
| PCML | $\mathbf{89.5}_{\pm2.0}$ |

## E.2  IMPLEMENTATION DETAILS

To determine the local manifold, we need to calculate the convex hull $\overline{U'}$ of latent representation set $U' = \{u'_j\}_{j=1}^n$. While the computation of the convex hull of a high-dimensional point set may be

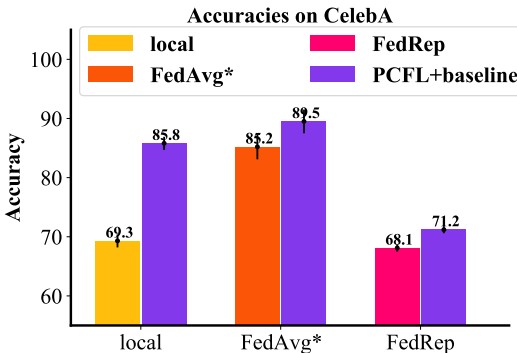

Figure 7: Visualization of the experimental results on CeleA dataset.

time-consuming. We approximate it with a minimum bounding box:

$$\{u \mid \min_{1 \leq j \leq n} u'_j|_k \leq u|_k \leq \max_{1 \leq j \leq n} u'_j|_k, 1 \leq k \leq d'\},$$

$u'_j|_k$ indicates the k-th element of the vector $u'_j$ and $d'$ is the dimension of the vector $u'_j$. The minimum bounding box encloses the convex hull $\overline{U'}$.

Our method is implemented with Pytorch and all experiments are run 5 times to calculate the average results with stds. We use a four-layer MLP for the synthetic experiment, three-layer MLP for FEM-NIST, two-layer CNNs for CIFAR10 and CelebA, and a one-layer MLP for Adult. Following the work (Collins et al., 2021), for all the methods we sample 10% of the clients in every global epoch. We train the models for 200 global epochs on FEMNIST, CIFAR10 and CelebA, 50 on Adult. And we train 15 local epochs for FEMNIST, CIFAR10 and Adult in every global epoch, 25 for CelebA. All models are trained with stochastic gradient descent. We use grid search to find the optimal hyperparameters $\alpha$ and $\beta$ in the validation set of each dataset. We set $\alpha = 0.5$, $\beta = 0.5$ for CIFAR10, CelebA, Adult; and set $\alpha = 1$, $\beta = 0.5$ for FEMNIST and eICU. Besides, we test different manifold dimensions $d'$ for each benchmark dataset. We keep $d'$ as small as possible while ensuring reconstruction quality on the validation set. We $d' = 256$ for CIFAR10 and CelebA, $d' = 12$ for FEMNIST, $d' = 32$ for Adult and eICU. For synthetic experiments, the data dimension $d = 3$ and manifold dimension $d' = 2$ since one element of data $z$ identically equals to 0. The source codes are made publically available at https://anonymous.4open.science/r/PCML-BACE/.

### E.3 DATASETS

In our experiments, CIFAR10, FEMNIST, CelebA and Adult are all public dataset. For the synthetic experiment, the data point $z = \{x, 0, y\}$ has three elements. We add a zero element to data so that the manifold dimension is smaller than the data dimension, which simulates the situation in real-world datasets. We create the collaboration environment with data heterogeneity for CIFAR10 and FEMNSIT by randomly allocating several classes to each client following the work (McMahan et al., 2017). For the dataset eICU, we follow the procedure on the website https://eicu-crd.mit.edu and got the approval for the dataset. We follow the data preprocessing as in Sheikhalishahi et al. (2019) and randomly select 14 hospitals as introduced in the main text.

### E.4 COMPUTING RESOURCES

Part of the experiments is conducted on a local server with Ubuntu 16.04 system. It has two physical CPU chips which are Intel(R) Xeon(R) CPU E5-2667 v4 @ 3.20GHz with 32 logical kernels. The other experiments are conducted on a remote server. It has 8 GPUs which are GeForce RTX 2080 Ti.

