# OpenReview forum: "Addressing Data Heterogeneity Through a Pre-learned Manifold for Distributed Learning Scenarios"
_ICLR.cc/2026/Conference — Submitted to ICLR 2026_

### Official Review · Reviewer_QJFR · 2025-10-27

**Soundness:** 3
**Presentation:** 2
**Contribution:** 2
**Rating:** 4
**Confidence:** 3

**Summary:**

This paper introduces the concept of Precision Collaboration, which applies the Pseudo-Invertible Encoder (PIE) approach [1,2] to address data heterogeneity in federated learning. Experiments on several popular benchmarks demonstrate its practical effectiveness.


[1] Beitler, Jan Jetze, Ivan Sosnovik, and Arnold Smeulders. "PIE: Pseudo-invertible encoder." arXiv preprint arXiv:2111.00619 (2021).
[2] Johann Brehmer and Kyle Cranmer. Flows for simultaneous manifold learning and density estimation. Advances in Neural Information Processing Systems, 33:442–453, 2020.

**Strengths:**

The proposed method is well-motivated and validated through synthetic experiments. And I appreciate the idea of drawing the samples from the unshared region and maximally use data from the shared region to improve local models' performance.

**Weaknesses:**

* The organization and presentation of contents make the paper hard to follow.
    * Several important details are deferred to the appendix, while others are omitted entirely (see questions below).
    * The paper introduces many mathematical notations, which is acceptable, but some are used without proper definition. This makes comprehension challenging on the first read — multiple passes are required to infer meaning from context. For example, the subscript $k$ in $z_k^j$ (equation 8) lacks clear explanations. The $\mathcal{F}$ in equation (9) and $d(.,.)$ in line 327 is not properly defined.

* Key components, such as learning the global manifold and estimating client-level densities over unshared data, appear to be direct extensions of PIE [1,2] to the federated learning setting. As a result, the paper’s contribution feels somewhat limited.


I am willing to raise scores if the questions are properly addressed, especially on the construction of $D^{i,j}$, the model architectures used to estimate the global manifold, and how to sample from $p_z^i$. Current ratings are negatively impacted by the key questions which I cannot find or infer the answers.

**Questions:**

* I think there could be a typo in equation (3). Since $e=g(z)$, I think the superscript $-1$ should be removed. Or $J_g$ should be $J_{g^{-1}}$. This issue also propogates to the density function (11) and the loss function in line-9 of Algorithm 1. Unfortunately, I could not verify this in the code because the `anonymous.4open.science` server was down during my review.

* The novelty of Proposition 1 is unclear. It appears to be a straightforward computation—Equation (16) in the proof is identical to Equation (3) in [2]. Moreover, from Equation (18), it is unclear how (19) follows, particularly how $c$ is determined to be a constant. Additionally, there may be an abuse of notation regarding
$p^{i}_z(z)$. Based on the context and the proof, the authors seem to be interested in derving the density function for the $i$-th client's data supported on the mainfold $\mathcal{M}_s^{i}$, i.e.,  the unshared region.

* How is $D^{i,j}$ deifned in the equation (8) constructed or estimated for the $i$-th client without sharing the latents $u'^{j}$ or the raw data $z^j$ from the $j$-th client.


* When learning the global data manifold, what are the architectures of $g_\theta$ and $h_{\phi}$? How do the authors ensure that they are diffeomorphisms? Again, I could not verify the code due to the inaccessible link.

* The authors mentioned that $\overline{U'^{i}}$ is approximated by the hyperrectangle. So the communication of $\overline{U'^{i}}$ reduces to $2d$ scalars. But how to approximate the $\overline{U'^{i}}$ and what is the communication costs of sending $\overline{U'^{i}}$ back to the client $i$?

* What is the computational cost of sampling from $p_z^i$ (the third term in equation (13))? If the constant $c$ (assuming the it's indeed a constant) is unknown, an MCMC-based approach may be required, which could be expensive.

* I would like to understand how the proposed method compare to the related methods like FedGen [3] in terms of the pros and cons and practical performances.

* Why do authors choose the PIE technique over the $\mathcal{M}$-flow and other variants mentioned in [2]?


[3] Zhuangdi Zhu, Junyuan Hong, and Jiayu Zhou. Data-free knowledge distillation for heterogeneous federated learning. arXiv preprint arXiv:2105.10056, 2021.

---

### Official Review · Reviewer_JxfQ · 2025-10-28

**Soundness:** 3
**Presentation:** 2
**Contribution:** 2
**Rating:** 2
**Confidence:** 4

**Summary:**

Paper proposes “Precision Collaboration for Model Learning (PCML)”: (i) learn a global normalizing-flow (NF) model to capture a “global manifold,” (ii) derive each client’s “local manifold,” (iii) detect inter-client overlaps in latent space (axis-aligned min–max box as a convex-hull proxy), and (iv) train each client on own data + overlapped samples + NF-generated samples. Experiments across synthetic, CIFAR-10, FEMNIST, Adult, eICU, and CelebA report gains over several FL baselines.

**Strengths:**

- Interesting “precision collaboration” idea: decouple representation (NF manifold learning) from downstream FL personalization; modular pipeline.
- NF enables explicit sampling likelihoods for targeted augmentation; plausible route to better coverage of rare/overlap regions.
- Explores both overlap-based sharing and generative augmentation; multi-dataset evaluation with 10- and 50-client variants.

**Weaknesses:**

- Writing clarity: The writing needs improvement and does not enumerate the pipeline, assumptions, privacy model, and computational footprint clearly; needs a crisp statement of “what we do” and “why it matters”, and "why it should work".
- Related work coverage: Needs a much deeper survey of clustered/inter- and intra-cluster sharing, representation/subspace sharing, and recent 2022–2024 personalized/clustered FL. Cited papers lean older; update to recent SOTA.
- Overlap assumption: The text states “there exist overlaps among data manifolds” (Line 204 p.4). This is not generally true; should be “may exist.” Many claims depend on actual overlap prevalence; add analysis of low/no-overlap regimes.
- Geometry→functional space linkage: The premise that latent-manifold proximity implies beneficial transfer in function space (nonlinear NNs) is asserted, not demonstrated. A shared geometric manifold need not yield similar optimal predictors; needs theory or targeted ablations.
- Overlap test looseness: Using an axis-aligned min–max “box” to approximate the convex hull in high dimensions is very loose, likely inflating false overlaps; no uncertainty or calibration shown. Consider tighter geometric tests (e.g., one-class boundaries/alpha shapes) and quantify FP/FN.
- Complexity: The manuscript suggests near O(1) overlap computation “once,” which is implausible given at least O(N)–O(N²) pairwise checks and dimensional dependence. Provide end-to-end complexity and wall-clock scaling vs clients/dimension, and update overlap costs during training.
- Privacy model: Clients transmit manifold summaries/U′ and a global NF is learned/shared, yet no formal privacy guarantees (DP, secure aggregation) or attack evaluations (membership/attribute inference) are provided. The paper acknowledges leakage risks but offers no mitigations/quantification. Privacy is so central in FL.
- “Global manifold” ambiguity: What exactly is the “global manifold,” which parameters are shared, and how is it learned while preserving privacy? Clarify threat model and communications.
- Algorithmic complexity/clarity: The pipeline has many steps (NF training, per-client manifold inference, overlap detection, sampling, multi-term local objective), multiple hyperparameters (α, β, thresholds), and deviates from standard FL training loops. The writing uses heavy notation, making Algorithm 2 hard to follow; add a simple schematic and sensitivity analysis.
- PACFL comparison (methodology and numerics): The approach is in spirit similar to PACFL (Vahidian et al., AAAI 2023), which communicates client subspaces/eigenvectors (U) to detect similarity/clustered collaboration. Here, PCML extracts manifolds (via NF) and ultimately also sends U-like summaries. Please add a thorough methodological comparison and head-to-head experiments against PACFL, with ablations isolating where PCML’s extra complexity helps. (Vahidian, Saeed, et al., AAAI 2023)
- Baselines and recency: Compared methods are mostly pre-2022; please include stronger modern SOTA personalized/clustered FL and representation-sharing baselines (2022–2024), plus fairness/graph-regularized FL where relevant.
- Experimental rigor/scope: FEMNIST is relatively easy; add harder datasets, deeper architectures, NLP tasks; vary clients widely; report seeds, CIs, and significance tests. Clarify hyperparameter tuning parity across methods; report ablations isolating contributions (overlap, sampling, NF capacity).
- Sensitivity: Performance appears dependent on hyperparameters (α, β, NF capacity, overlap thresholds). Provide robustness studies, stability across seeds, and convergence diagnostics.
- Convergence: No convergence theory or even empirical convergence diagnostics; include at least training dynamics and failure modes.
- Presentation/typos/notation: Several typos (“embeded,” “likelyhood,” “CeleA,” FEMNSIT) and inconsistent notation (y-hat placement); some figure legends mislabel (e.g., “PCFL+baseline”). Figures lack complete settings and error bars. These issues hamper readability and reproducibility.

**Questions:**

1) Conflict and benefit: Under what conditions does manifold proximity guarantee improved function-space generalization? Provide theory (bounds or assumptions) or diagnostics (e.g., gradient-conflict measures, NTK similarity, Jacobian overlap).
2) Overlap detection: Why min–max boxes instead of tighter detectors? Quantify false overlaps and their effect on utility/privacy; add uncertainty-aware thresholds.
3) Complexity and comms: What is the precise runtime/communication complexity vs clients (N), overlap checks, and latent dimension? Report scaling plots and update-costs across rounds.
4) Privacy: Define the threat model; evaluate membership/attribute inference on shared U′/NF params; optionally add a DP variant and report the privacy–utility trade-off.
5) PACFL: Provide a clear methodological comparison to PACFL (principal angles between client data subspaces) and comprehensive empirical comparisons; discuss when PCML’s manifold machinery outperforms/underperforms PACFL and why (cost vs benefit).
6) Robustness: Add sensitivity to α, β, NF depth/width, latent dimension d′, overlap thresholds; add seed CIs and per-client variance; report failure/edge cases (no/rare overlaps).
7) Reproducibility: Specify hyperparameter search space, seeds, budgets, and ensure tuning parity across baselines; define the hypervolume/reference points if used; include error bars.

---

### Official Review · Reviewer_9AuE · 2025-10-29

**Soundness:** 3
**Presentation:** 3
**Contribution:** 2
**Rating:** 6
**Confidence:** 3

**Summary:**

This paper addresses the challenge of data heterogeneity in distributed learning frameworks such as Federated Learning (FL). The authors argue that the limited information-sharing inherent in collaborative networks undermines the assumptions made in prior works. To explain and formalize this issue, they introduce the concept of Precision Collaboration and further propose leveraging a pre-learned global manifold as a potential solution. The overall logic of the paper is coherent and well-structured, providing a clear rationale for the proposed approach.

**Strengths:**

-The paper provides a thoughtful discussion of the fundamental mechanisms underlying heterogeneity issues in decentralized learning and effectively highlights a key limitation in the assumptions made by previous studies. This conceptual analysis adds valuable insight to the understanding of heterogeneity in distributed learning systems.

-The paper introduces the Precision Collaboration concept to characterize and explain the information-sharing phenomenon in collaborative networks. The accompanying discussion on fragmented distribution overlaps is insightful and provides an inspiring perspective on how data heterogeneity affects collaboration dynamics in decentralized learning.

**Weaknesses:**

-While the paper’s title and introduction emphasize distributed learning, the experimental section is limited to FL scenarios. Since FL is a specific subset of distributed learning that relies on a central server, the scope of the experiments does not fully support the broader claims made in the title. If the evaluation focuses solely on FL, it would be more appropriate to frame the work explicitly within the federated learning context rather than the more general distributed learning paradigm.

-The proposed method appears somewhat trivial, as it primarily relies on the existing NF tool without substantial methodological innovation or adaptation. Moreover, the effectiveness of the approach seems to depend heavily on acquiring data from all clients, which raises concerns about its practicality and alignment with real federated or decentralized learning constraints.

**Questions:**

-It remains confusing how the global manifold is trained in a FL setting. Since direct access to users’ data is strictly prohibited, the assumption of having access to all client data for training the manifold appears unrealistic in real-world scenarios. Could the authors clarify how the global manifold is learned while adhering to the privacy and data isolation constraints of FL?

-What would happen if newly joined clients possess manifolds that differ from the learned global manifold, which is a highly likely scenario in practice? There is inherently a trade-off between adapting to the known data distribution and generalizing to unseen or shifted distributions. How does the proposed method handle this trade-off, and can it effectively generalize to clients with substantially different underlying manifolds?

---

### Official Review · Reviewer_sVQA · 2025-11-13

**Soundness:** 3
**Presentation:** 3
**Contribution:** 3
**Rating:** 4
**Confidence:** 4

**Summary:**

Data heterogenity is a situation in distributed training (mostly federated learning) where the local datasets on different clients have considerably different data/label distributions. It is a problem because it causes client models to diverge, as each client trains on a subset of the data that is out-of-distribution for other clients. A fundamental question when designing a practical federated learning system with heterogenous data is whether we should use a single (global) model that does its best to serve each client, or whether we should learn a unique (personalized / local) model that is optimized for each client. When we operate in the first setting, we wish to maximize performance over the entire data distribution and are typically concerned with problems like training instability and convergence, since we must learn a global model from what is effectively a dataset of poorly-shuffled, biased batches. When we operate in the second setting, we are interested in maximizing performance on a per-client basis. Here, we usually try to identify clients whose data distributions can be combined in some way to yield a better model than would be possible using a single client alone (through more data, better generalization, etc).

This paper proposes a technique for the personalized setting called "Precision Collaboration," which enables a client model to find other client models with overlapping distributions, and learn from them by (a) training on the subset of the other client's data that overlaps with the local distribution and (b) generating data that matches the local distribution model. Precision collaboration does this by (1) fitting a (generative) distribution model on the merged data from all clients as well as locally on each client, (2) using the distribution models to determine which data points overlap for each pair of clients, (3) allowing clients to train on the overlapping points, and (4) using the distribution model to generate synthetic examples when there is no overlapping client from which to mine extra points. There is a detailed exposition accompanied by a good set of experiments in federated learning (FL) that demonstrate performance benefits from precision collaboration.

**Strengths:**

1. It is an interesting approach to share information between clients at the sub-client-dataset level, rather than the more standard client-cluster level.
2. The experiments are done well and compare against a sufficiently large number of baselines as to inspire confidence in the method.
3. The paper is well-written and easy to follow.

**Weaknesses:**

1. **Privacy:** In FL we typically want rigorous, provable notions of privacy - usually differential privacy. The manifold model training process described in Section 4 does not adhere to DP (though I imagine that it may not be hard to make it do so, e.g. through the use of differentially-private gradients). Some of the baselines in the experiment section do accommodate DP so I am not sure whether the experiments do the right thing and compare against a non-private version of baselines (for an equivalent point on the privacy-utility tradeoff).
2. **Details:** The main text seems a little light on details (it likely just needs to add references to the appendix). For example, in the main text I could not find pointers or descriptions of the network architectures used in the experiments, the architecture / size of the manifold model, the method used to determine whether a distribution overlap occurred, and how to combine the synthetic samples from the manifold model, the contributions from the overlap components of other clients, and the local data in the objective.
3. **Literature context:** The paper uses a parametric model to learn the data manifold, which is part of the reason why privacy is tricky (we have to learn the data model). However, there are a lot of recent clustered FL methods, some of which model the distribution using non-parametric (and differentially-private) models, and many of these are not discussed. For example, in the paper "One-Pass Distribution Sketch for Measuring Data Heterogeneity in Federated Learning" at NeurIPS'23, the authors use a private density model to estimate the amount of overlap between client data distributions. In "Optimizing the collaboration structure in cross-silo federated learning" at ICML'23, the authors do basically the same thing but this time using the C-divergence. There is a benchmarking paper ("Benchmarking Data Heterogeneity Evaluation Approaches for Personalized Federated Learning") that goes through a number of options to estimate data overlap in the FL setting. Even if these methods are not represented as baselines in the experiments, I still think that the paper should discuss these prior efforts to quantify heterogenity and use that measurement in the learning algorithm.
4. **Overlap method:** My understanding is that you apply a trained encoder to get a latent representation for each point in the local data. Then, these points are passed to a convex-hull algorithm, which we call $O(N^2)$ times to determine the points which "overlap" between each pair of clients. This procedure seems like it would be very sensitive to outliers in the local client data, because adding a single point could cause the overlap to increase arbitrarily. Am I missing something here? It would be nice to validate that the overlap estimation method is actually following our intuition, since it seems possible to break it with a single adversarial example.

Plus a typo:
- "In particular, Given the data" -> "In particular, given the data" on L228

**Questions:**

Aside from addressing the weaknesses, I have the following specific questions:
1. What is the architecture of the data manifold model? Is it a variational autoencoder or similar?
2. Is there any ablation study measuring the performance contributions of the sampled data and the overlaps?
3. Is there any easy way to make this have privacy guarantees? Also, are the experiments conducted with apples-to-apples privacy-utility tradeoff settings?
4. Have you considered more robust measures of data overlap than convex hull?

---

### Meta-Review · Area_Chair_2NS9 · 2026-01-04

**Summary:**

This paper introduces "Precision Collaboration", a method that leverages a pre-learned normalizing flow manifold to address data heterogeneity in FL by identifying overlapping data distributions across clients.

I summarize the key concerns below:
- **Privacy**: It is unclear how the global manifold is trained within FL constraints without accessing raw client data. No formal privacy guarantees or empirical tests are provided.
- **Presentation**: Critical details are missing or relegated to appendices. The notation is heavy and at times inconsistent. These issues prevented reviewers from fully evaluating the technical claims.
- **Novelty**: Multiple reviewers noted the approach appears to be a relatively straightforward application of existing normalizing flow techniques to the FL setting, without substantial methodological innovation.
- **Experimental Rigor:** The paper lacks clarity on computational and communication complexity. Missing or unclear hyperparameter search space, seeds, budgets, and tuning of the baselines. The comparison with the related method PACFL is also insufficient.

None of the authors championed this paper, and the fundamental issues in privacy, novelty, and rigor -- shared by all the reviewers -- reflect consensus that the paper falls below the acceptance threshold.

**Reviewer Concerns:**

No rebuttal provided.

**Reviewer Scores:**

No rebuttal provided.

---

### Decision · Program_Chairs · 2026-01-26

Reject